# Does Cardiac Function Affect Cerebral Blood Flow Regulation?

**DOI:** 10.3390/jcm11206043

**Published:** 2022-10-13

**Authors:** Shigehiko Ogoh, Jun Sugawara, Shigeki Shibata

**Affiliations:** 1Department of Biomedical Engineering, Toyo University, 2100 Kujirai, Kawagoe-shi 350-8585, Japan; 2Neurovascular Research Laboratory, Faculty of Life Sciences and Education, University of South Wales, Pontypridd CF37 4AT, UK; 3Human Informatics and Interaction Research Institute, National Institute of Advanced Industrial Science and Technology, Tsukuba 305-8566, Japan; 4Faculty of Health and Sport Sciences, University of Tsukuba, Tsukuba 305-8574, Japan; 5Department of Physical Therapy, Faculty of Health Science, Kyorin University, Tokyo 181-8611, Japan

**Keywords:** heart failure, cardiac dysfunction, cerebral perfusion, cognitive function

## Abstract

Many previous studies indicate that heart failure (HF) increases the risk of cognitive dysfunction and stroke, showing the logic that several physiological factors associated with cardiac dysfunctions affect homeostasis in the cerebral circulation. In the chronic process of HF patients, it is suggested that reduced cerebral blood flow (CBF) and abnormal auto-regulation might result in impaired perfusion, metabolic insufficiency, and regional or global structural deteriorations in the brain. However, the mechanism underlying HF-induced brain disease remains unclear. Cardiac dysfunction in patients with HF or HF-induced several physiological abnormalities may cause brain dysfunction. Possible physiological factors should be considered for future studies to prevent brain disease as well as cardiovascular dysfunction in patients with HF.

## 1. Introduction

It has been reported that coronary heart disease almost triples the risk of a stroke, and cardiac failure is associated with a more than fivefold increased risk [1]. In the chronic process of heart failure (HF) patients, it is suggested that reduced cerebral blood flow (CBF) and abnormal auto-regulation might result in impaired perfusion, metabolic insufficiency, and regional or global structural deteriorations in the brain [2,3] (Table 1). The physiological mechanism by which heart disease increases the risk of brain disease remains unknown; however, previous reports suggest that several physiological factors associated with cardiac dysfunctions in patients with HF directly affect homeostasis in the cerebral circulation. From these previous studies, the influence of physiological factors regarding HF on cerebral circulatory homeostasis ought to be considered. In this review, regarding the high risk of brain disease in patients with HF, we would like to focus on the physiological mechanisms of CBF regulation altered by HF to prevent brain disease or identify adequate clinical treatment in patients with HF.

## 2. Does Cardiac Dysfunction Determine Characteristics in Cerebral Circulation?

HF, one of the leading causes of morbidity and mortality, is a systemic disease but affects the other organ system, especially the brain [13]. For example, it has been reported that as many as 35% to 50% of patients with HF have cognitive dysfunction and cognitive impairment, which are associated with increased disability and mortality in patients with HF [14]. Additionally, many previous studies indicate that HF increases the risk of cognitive dysfunction and stroke [15,16,17]. Therefore, these findings support the logic that several physiological factors associated with cardiac dysfunctions affect homeostasis in the cerebral circulation; however, the mechanism underlying HF-induced brain disease remains unclear. Its several clinical direct and indirect pathophysiologic mechanisms have been described [15], and it is thought that the direct factors of cardiac dysfunction against the brain are HF-induced left ventricular (LV) hypokinesia [18,19], hypoperfusion [20] or hypotension [21], while the indirect factors are an impairment in cerebral autoregulation [4], endothelial dysfunction [22], and so on.

In severe chronic HF patients, global CBF is reduced by 14–30% [23]. Kim et al. [13] found that HF patients with low CBF were nearly 2.5 times more likely to die or require urgent transplantation during a median follow-up period of 3 years, and suggested that CBF measurement is important as a promising prognostic tool to identify advanced HF patients. Thus, cerebral hypoperfusion may be key to causing brain disease in HF patients. The relationship between the changes in CBF velocity (MCA) and the changes in cardiac output at rest and during dynamic exercise was linear and highly significant [24], indicating that carotid–cardiac baroreflex function or cardiac function-mediated cardiac output, as well as blood pressure, is important to the regulation of CBF velocity [25]. Indeed, a decrease in the response of cardiac output to exercise associated with cardiac dysfunction (beta blocker [26], and arterial fibrillation [6]) attenuates the exercise-induced increase in CBF velocity. The recent treatment strategies for HF patients are pharmacological therapies such as diuretics, beta blockers, renin–angiotensin system inhibition with Angiotensin Converting Enzyme Inhibitors (ACEi) or Angiotensin (II) receptor blockers (ARB) or Angiotensin Receptor-Neprilysin Inhibitor (ARNi), device therapies, and interventional therapies [27]. Among them, HF-induced low CBF is reported to be recovered after ACEi treatment [23], cardioversion [28], cardiac resynchronization therapy [29], and heart transplantation [30]. Thus, under these backgrounds, it can be speculated that HF-induced decrease in cardiac output and hypotension may decrease CBF via cardiac dysfunction.

On the other hand, Choi et al. [2] reported that global CBF at rest was decreased in advanced HF (according to NYHA); however, they demonstrated that the decrease in CBF was not dependent on cardiac dysfunction in the patients with HF, i.e., exercise capacity (heart volume) or left ventricular ejection fraction (LVEF, cardiac function). Thus, the decrease in CBF in HF patients may be caused by indirect HF-induced physiological factors rather than direct cardiac dysfunction. The unrelatedness of cardiac dysfunction and abnormal CBF response may be related to compensatory mechanisms against cardiac dysfunction-induced abnormal perfusion (cardiac output) or perfusion pressure (blood pressure) to maintain an adequate CBF.

## 3. Conceivable “Compensatory Mechanism” for Alterations in Cardiac Output and Arterial Blood Pressure to Maintain an Adequate CBF in Patients with HF

Previous studies demonstrated some compensatory mechanisms (cerebral autoregulation, cardiac baroreflex, cardiopulmonary baroreflex, external carotid artery) against changes in arterial pressure and cardiac output to maintain adequate CBF [31]. Thus, it is possible that these compensatory mechanisms are altered in patients with HF and consequently modified CBF regulation.

### 3.1. Cerebral Autoregulation

Cerebral autoregulation, which maintains steady-state cerebral blood flow in a relatively stable manner over a range of perfusion pressures from 60 to 150 mmHg [32,33], is an essential physiological mechanism of cerebral vasculature to maintain an adequate CBF. The concept of cerebral autoregulation is important to determine steady-state CBF but it is noteworthy that it takes ~3 s for dynamic cerebral autoregulation to be established [34,35]. Thus, the CBF in basal cerebral arteries fluctuates in parallel with the change in blood pressure throughout the cardiac cycle. Recently, it has been reported that patients with chronic HF mainly due to aortic valve disease have a significantly higher dynamic cerebral autoregulation [5]. This finding indicates that cerebral autoregulation is intact even in patients with HF, and cerebral autoregulation works to compensate for HF-induced hypotension for CBF regulation. In other words, enhanced dynamic cerebral autoregulation is essential for cerebral hypotension in patients with HF. Thus, the influence of chronic antihypertensive medication on the dynamic cerebral autoregulation status of HF patients needs to be considered carefully because it is possible that these drugs modify sympathetic activity that alters cerebrovascular regulation [36,37].

### 3.2. Arterial Baroreflex Function

Arterial baroreflex function is the compensatory mechanism for acute changes in arterial pressure to maintain an adequate perfusion pressure. Ogoh et al. [25] demonstrated that tachycardia response via cardiac arterial baroreflex function contributes to dynamic cerebral autoregulation during acute hypotension, indicating that cardiac arterial baroreflex plays an essential role in dynamic CBF regulation. It has been reported that cardiac arterial baroreflex sensitivity is attenuated in HF patients [7], but HF enhances cerebral autoregulation [5] rather than attenuates it. Thus, the effect of HF-induced attenuation in arterial baroreflex function that contributes to cerebral autoregulation on CBF regulation may be minimal. However, it cannot be ruled out that an impairment in arterial baroreflex in patients with HF may be associated with HF-induced hypotension and consequently low CBF. On the other hand, it is possible that enhanced cerebral autoregulation may compensate for attenuation in arterial baroreflex function-induced hypotension. Continuous-flow left-ventricular-assist devices (LVAD) improved outcomes including autonomic tone for patients with advanced HF; however, the reduction in cardiac baroreceptor sensitivity persisted [7]. Moreover, exercise-induced increases in arterial blood pressure are blunted, indicating that arterial blood pressure regulation is still attenuated, although cardiac function recovered. This clinical treatment could not recover the attenuation in arterial baroreflex and thus it is possible that patients with HF cannot reduce the risk of brain disease via this treatment. Treatment with beta blockers reduces the risk of death and the combined risk of death or hospitalization in patients with HF with reduced ejection fraction (HFrEF), and thus it is widely used in HFrEF patients [27]. While it is known that beta blockers improve LVEF, their effects on baroreflex function through the autonomic nervous system may affect brain autoregulation. However, previous physiological studies [38,39] showed that beta blockers did not impair arterial–cardiac baroreflex at rest and during exercise, while vagal blockade did impair it. Thus, beta blockers may have beneficial effects on cerebral autoregulation, especially in patients with LVEF improved by beta blockers. More recently, ivabradine was suggested to be used for HF patients with sinus rhythm and a heart rate of ≥70 bpm at rest, in order to reduce hospitalizations and cardiovascular death [27]. Recent animal studies indicated that ivabradine affects neither cardiovascular autonomic control nor arteria baroreflex function [40,41]. Thus, heart rate control therapies by these drugs are not likely to impact cerebral autoregulation via baroreflex function.

### 3.3. Cardiopulmonary Baroreflex

In addition to arterial baroreflex, the cardiopulmonary baroreflex is also essential for CBF regulation. An increase or decrease in central blood volume (CBV) possibly elevates or reduces cardiac output and consequently causes over- and under-cerebral perfusion [24]. Orthostatic stress decreases CBV but then cardiopulmonary baroreceptors are unloaded and elevate peripheral vascular resistance via an increase in sympathetic nerve activity to prevent a large reduction in CBF [42,43,44]. On the other hand, attenuated sympathetic response via cardiopulmonary baroreflex to orthostatic stress easily causes syncope [45] because of a large decrease in CBF. In contrast, an increase in central blood volume, i.e., microgravity [46], or onset of cycling via muscle pump [47] loads cardiopulmonary baroreceptors and consequently decreases peripheral vascular resistance to prevent over-cerebral perfusion. Therefore, cardiopulmonary baroreflex associated with central blood volume is an essential indirect CBF regulatory mechanism. Cardiopulmonary baroreflex as well as arterial baroreflex is reported to be impaired in patients with severe HF [8,9,10,48]. A reduced cardiopulmonary baroreflex has been already detected in the early phase of HF [11]. This finding indicates that HF-induced low CBF is partly determined by an impairment in cardiopulmonary baroreflex. In patients with HF who have fluid retention, diuretics such as loop and thiazide are recommended to relieve congestion, improve symptoms, and prevent worsening HF [27]. Since these drugs aim at blood volume reduction to improve congestion, they may increase the risk of severe cerebral hypo-perfusion given that cardiopulmonary baroreflex is impaired in HF patients. However, since the treatment effects of these drugs on cardiopulmonary baroreflex itself are still unknown, further studies would be warranted. Renin–angiotensin system inhibition with ACEi, ARB, or ARNi may also affect cardiopulmonary baroreflex by reducing blood volume and vasoconstriction. Physiologically, this treatment directly exacerbates cardiopulmonary baroreflex through impaired vasoconstrictive ability via blockade of the renin–angiotensin system, leading to worse CBF regulation. However, one previous study suggested that the treatment of HF patients by ACEi improved the outflow of sympathetic nerve activity in the cardiopulmonary baroreflex pathway [49]. The gross effects of renin–angiotensin system inhibition on CBF regulation are still unknown and future studies are warranted.

### 3.4. External Carotid Artery

Moreover, the vascular bed of the external carotid artery is essential to regulate intracranial blood flow. For example, resistance-exercise-induced large arterial blood pressure increases common carotid artery blood flow but internal carotid artery blood flow that supplies blood to the cerebral cortical region is well maintained, while external carotid artery blood flow largely increased accordingly as a large increase in arterial blood pressure [50]. This finding indicates that, in the cerebral circulation, the vascular bed of the external carotid artery buffers acute increase in CBF, similar to the Windkessel model in the peripheral vasculature. Indeed, intracranial pressure interacts with an increase in external carotid artery blood flow to prevent an increase in CBF [51]. In contrast, external carotid artery blood flow decreases during an acute hypotension-induced decrease in common carotid artery blood flow to supply blood to the intracranial cerebral artery [52]. Moreover, augmented cardiac contractility using dobutamine increases cardiac output but increases external carotid artery blood flow to prevent an increase in internal carotid artery blood flow [53]. These findings strongly suggest that the vascular bed of the external carotid artery is an important buffering mechanism for preventing over-or hypo-perfusion in the intracranial cerebral anterior artery. However, so far, it is unclear whether this external vascular bed compensatory mechanism for cardiac dysfunction works in patients with HF. Stiffing this vascular bed should attenuate this compensatory mechanism to buffer changes in cardiac output and perfusion pressure, but we need further investigation if HF patients alter characteristics in the external carotid artery.

### 3.5. Neurohormonal Systems of HF

It is known that sympathetic nerve activity and the renin–angiotensin-aldosterone system are enhanced in HF patients [54]. The compensatory homeostatic responses to a fall in cardiac output are activation of the sympathetic nervous system and the renin–angiotensin-aldosterone system. This neurohormonal activation is likely beneficial for preserving the blood flow of the brain and other organs in the acute phase of HF. However, with chronic activation, these responses may result in deteriorated effects on the cardiovascular function and morphology including cerebrovascular tissues, possibly leading to reduced CBF in chronic HF patients. Thus, paradoxically, neurohormonal activation to preserve CBF in chronic HF patients is recognized as the most important pathophysiology underlying the progression of HF. Consequently, current pharmacological therapies are targeted to block these neurohormonal activities, in addition to diuretics. However, the direct effects of neurohormonal systems on cerebral circulation and their treatment effects by drugs are still unknown in chronic HF patients, as described above.

## 4. Possible other Physiological Factors for Impaired Cerebral Circulation in Patients with HF

As well as the CBF compensatory mechanism, we may need to consider possible HF-induced physiological factors that decrease CBF rather than cardiac dysfunction in HF patients. One of them is, for example, greater arterial stiffness that is associated with left ventricular (LV) remodeling and reduced LV function [55,56]. HF patients with a preserved ejection fraction (HFpEF), exhibit higher aortic stiffness, reflecting the impaired Windkessel function [12,57]. Central elastic arteries that connect the heart to the brain (e.g., aorta and carotid artery) expand with a cyclic cardiac ejection as they accommodate part of the blood inside during the systole and then recoil and transmit the pooled blood to the periphery, including the brain tissue, during the diastole. The impairment of this function results in not only exaggerated increases in central systolic blood pressure and pulse pressure but also drops in blood pressure and CBF in diastole [58]. In turn, increased arterial stiffness contributes to the brain’s chronic exposure to cyclic mechanical forces of cardiac pulsations because the brain with very low vascular resistance is continually and passively perfused at a high-volume flow throughout the systole and the diastole. It may promote abnormalities in the microvascular structure and function [59]. We confirmed in healthy individuals with a broad age range (21–79 tears) that central arterial stiffness is positively associated with cerebrovascular impedance (an index of dynamic cerebrovascular tone) and that higher cerebrovascular impedance results in brain hypoperfusion [60]. These findings suggest that increases in cerebrovascular impedance may attenuate systemic arterial pressure pulsatility at the cost of increased risks of brain hypoperfusion. Cardiac dysfunction and cerebral unfavorable circulatory homeostasis may be linked by a common factor: central arterial stiffening. These findings suggest that arterial stiffness is associated with alterations in CBF regulation, and thus it is possible that HF-induced arterial stiffness causes cerebral hypoperfusion.

## 5. Conclusions

Based on the anatomy, cardiac function should directly affect CBF regulation because of the cerebral artery connected to the aortic arch. Indeed, changes in arterial blood pressure and cardiac output are directly associated with CBF regulation. However, cardiac dysfunction in patients with HF directly does not cause impairment in CBF regulation. The compensatory mechanism of CBF regulation; cerebral autoregulation, baroreflexes, and external carotid artery vascular bed, against cardiac dysfunction, are also modified in patients with HF. Thus, these alterations may affect CBF regulation. However, since sympathetic nerve activity and the renin–angiotensin–aldosterone system that play a role in blood volume regulation are enhanced in HF patients, the effect of these physiological factors on the compensatory mechanism of CBF regulation needs to be considered. Additionally, one of the possible important physiological factors of cerebral hypoperfusion in patients with HF is arterial stiffness. Our previous study demonstrated that arterial stiffening modified dynamic pulsatile hemodynamic transmission from the aorta to the brain. Cardiac dysfunction and cerebral unfavorable circulatory homeostasis may be linked by a common factor: central arterial stiffening rather than direct cardiac dysfunction.

## Figures and Tables

**Table 1 jcm-11-06043-t001:** Physiological factors, which as the risk of brain disease, altered in patients with Heart Failure (HF) reported in human studies.

Author	Year	Region	Subject				Type of Study	Physiological Factor	Response in Patients with HF
			HF Patients	Number	Control	Number			
**Cerebral Blood Flow**									
Choi et al. [2]	2006	Korea	Advanced HF secondary to idiopathic dilated cardiomyopathy (ejection fraction ≤ 35%)	52 (41 ± 11yrs.)	Healthy	10 (39 ± 13 yrs.)	Comparison study (HF vs. Control)	Global CBF	Decreased (−19%, *p* < 0.001)
Roy et al. [3]	2017	USA	HF with hypertensive (n = 12), atrial fibrillation (n = 4), and a history of type 2 diabetes (n = 5). NYHA functional class II (80%) and III (20%)	19 (56 ± 9 yrs.)	Healthy	29 (51 ± 5 yrs.)	Comparison study (HF vs. Control)	Regional CBF	Decreased (MRI data, hippocampus, anterior thalamus, occipital cortex etc., *p* > 0.004)
**The Regulation of Cerebral Vasculature**									
Georgiadis et al. [4]	2000	Italy	NYHA II (n = 19), NYHA III (n = 21), and NYHA IV (n = 10)	50 (61~53 yrs.)	Healthy	20 (57 ± 9 yrs.)	Comparison study (HF vs. Control)	Cerebrovascular reactivity (CVR)	Decreased (Linear relationship between LVEF and CVR, r^2^ = 0.21, *p* < 0.001)
Castro et al. [5]	2020	Portugal	Stroke Patients with HF (n = 8) and Non-HF (n = 42)	50 (73 ± 12 yrs.)	Healthy	50 (71 ± 6 yrs.)	Comparison study (HF vs. Control)	Dynamic cerebral autoregulation	Increased (TFA phase, HF compared with Non-HF, *p* < 0.001)
**Cardiac Output**									
Ide et al. [6]	1999	Denmark	Atrial Fibrillation	11 (69, 64~73 yrs.)	Healthy	5 (64, 61~70 yrs.)	Comparison study (HF vs. Control)	Cerebral blood flow and cardiac output during exercise	There is a correlation between the increase in CBF and the ability to increase cardiac output (r^2^ = 0.55, *p* < 0.01).
**Carotid Baroreflex**									
Sailer et al. [7]	2021	USA	Patients with advanced HFrEF and were scheduled to undergo CF-LVAD implantation	12 (60 ± 11 yrs.)	n/a	n/a	Comparison study (Pre vs. Post left ventricular assist devices, LVAD)	Cardiac baroreflex	Continuous flow-LVAD implantation is associated with modest improvements in autonomic tone, but persistent reductions in cardiac baroreceptor sensitivity and BP response is blunted
**Cardiopulmonary baroreflex**									
Ferguson et al. [8]	1984	USA	NYHA II (n = 4), NYHA III (n = 5), NYHA IV (n = 2)	11 (37 ± 4 yrs.)	Healthy	17 (25 ± 1 yrs.)	Comparison study (HF vs. Control)	Cardiopulmonary baroreflex	Attenuated (Forearm vascular response to orthostatic stress, *p* < 0.01)
Mohanty et al. [9]	1989	USA	NYHA III and IV	29	Healthy	11	Comparison study (HF vs. Control)	Cardiopulmonary baroreflex	Attenuated (Forearm vascular response to orthostatic stress)
Creager et al. [10]	1990	USA	NYHA III (n = 7), IV (N = 5)	12 (63 ± 9 yrs.)	Healthy	20 (26 ± 5 yrs., published as a separate report, AJP H219-25, 1989)	Comparison study (HF vs. Control)	Cardiopulmonary baroreflex	Baroreceptors can regulate splanchnic and renal but not limb vascular resistance
Modesti et al. [11]	2004	Italy	NYHA I (n = 18), NYHA II (n = 13)	31 (67 ± 10 yrs.)	Healthy	11 (53 ± 10 yrs.)	Comparison study (HF vs. Control)	Cardiopulmonary baroreflex	Reduced only in NYHA II patients (*p* < 0.03) but not in NYHA I.
**Arterial Stiffness**									
Mottram et al. [12]	2005	Australia	Hypertensive patients with suspected diastolic HF	70	Healthy	15	Comparison study (HF vs. Control)	Arterial compliance	Attenuated (*p* = 0.011)

## Data Availability

Not applicable.

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
