# Peer review of "Does Cardiac Function Affect Cerebral Blood Flow Regulation?"

_jcm, 2022, doi:10.3390/jcm11206043_

Round 1

Reviewer 1 Report

The authors investigated whether cardiac function affects cerebral blood flow regulation. They concluded that, cardiac function should directly affect CBF regulation because of the cerebral artery connected to the aortic arch.

I have the following concerns:

1. The included studies should be listed in table with number of patients, region, and design.

2. What are the practical implications of the study?

3. Were patients with HFpEF included?

Author Response

Response to reviewers

For Reviewer 1

Comments and Suggestions for Authors

The authors investigated whether cardiac function affects cerebral blood flow regulation. They concluded that, cardiac function should directly affect CBF regulation because of the cerebral artery connected to the aortic arch.

REPLY: We appreciate the reviewer’s supportive comments and constructive suggestions. We have considered the comments respectfully and carefully and inserted our responses to each comment below. We hope that this revision is improved and that you are satisfied with the changes made.

I have the following concerns:

  1. The included studies should be listed in table with number of patients, region, and design.

REPLY: Thank you so much for your constructive suggestion. As you suggested, we have added a new table to be clear for readers (P2).

  1. What are the practical implications of the study?

REPLY: We apologize for unclear the practical implications of the study. In this review, regarding the high risk of brain disease in patients with HF, we would like to focus on the contribution of physiological mechanisms (new aspect for CBF regulation) altered by HF to CBF regulation to prevent brain disease or identify adequate clinical treatment in patients with HF. As you suggested, we have added the practical implications of this review in the text to be clear for readers (P1 L36-39).

We have added

“In this review, regarding the high risk of brain disease in patients with HF, we would like to focus on the physiological mechanisms of CBF regulation altered by HF to prevent brain disease or identify adequate clinical treatment in patients with HF.”

  1. Were patients with HFpEF included?

REPLY: Thank you for your comments. Yes, we have already discussed HFpEF with arterial stiffness in patients with HF in the part (4. Possible other physiological factors for impaired cerebral circulation in patients with HF) of the transition from the systemic circulation to the cerebral circulation (P6 L246-256).

Reviewer 2 Report

The authors reviewed the relationship between cardiac dysfunction and cerebral blood perfusion. The theme of the article was interesting and clinically relevant, however, I have several concerns.

Major concerns

#1

Majority of the references are out-of-date. Treatment strategy and prognosis in patients with heart failure have changed significantly in this decade.

#2

It seems that the authors recognized heart failure as a mechanical disease, consists of cardiac output, blood volume, vascular resistance, and so on. However, particularly in chronic heart failure, the disease should be recognized as a result of complex interaction between mechanical circulatory systems and several neurohormonal systems, including sympathetic nerve systems and renin-angiotensin-aldosterone system. These neurohormonal systems also affect the cerebral blood flow, however, discussion regarding these systems seems lacking.

#3

In each paragraph, the authors presented several papers regarding some specific cardiac function and cerebral blood flow. After the review of these papers, the authors stated their opinion regarding each specific relationship. For example, they used “may be due to (line 120)” or “clearly (line 167). However, these notions seem not always follow the results of the papers cited, rather, reflect the opinions of the authors. It is recommended the authors should discriminate their opinion from the results of the references.

Minor concerns

#1

Regarding lines 92-99,.

The authors stated arterial baroreflex function initiate tachycardia as a response to reduced cardiac function. In general, higher heart rate is a worse prognostic indicator for heart failure. In addition, beta-blockers or ivabradine, both make heart rate slow,  are widely used in patients with heart failure with reduced ejection fraction. Please discuss the effect of these drugs on the cerebral blood perfusion.

#2

Regarding lines 105-121

The authors discussed the effect of cardiopulmonary baroreflex on cerebral blood flow.

The component of cardiopulmonary baroreflex, blood volume and peripheral vascular resistance, are affected significantly by treatment for heart failure, i.e. diuretics or renin-angiotensin-aldosterone system inhibitors. Please discuss the role of these drugs in regard to cardiopulmonary baroreflex.

Author Response

For Reviwer2

Comments and Suggestions for Authors

The authors reviewed the relationship between cardiac dysfunction and cerebral blood perfusion. The theme of the article was interesting and clinically relevant, however, I have several concerns.

REPLY: We appreciate the reviewer’s supportive comments and constructive suggestions. We have considered the comments respectfully and carefully and inserted our responses to each comment below. We hope that this revision is improved and that you are satisfied with the changes made.

Major concerns

#1 Majority of the references are out-of-date. Treatment strategy and prognosis in patients with heart failure have changed significantly in this decade.

REPLY: Thank you so much for your constructive comments. As you suggested, we have rewritten the up-date treatment strategy and prognosis in patients with heart failure (P3 L74-82).

We have added

“The recent treatment strategies for HF patients are pharmacological therapies such as diuretics, beta-blockers, renin-angiotensin system inhibition with Angiotensin Converting Enzyme Inhibitors (ACEi) or Angiotensin (II) receptor blockers (ARB) or Angiotensin Receptor-Neprilysin Inhibitor (ARNi), etc., device therapies, and interventional therapies [20]. Among them, … “

#2 It seems that the authors recognized heart failure as a mechanical disease, consists of cardiac output, blood volume, vascular resistance, and so on. However, particularly in chronic heart failure, the disease should be recognized as a result of complex interaction between mechanical circulatory systems and several neurohormonal systems, including sympathetic nerve systems and renin-angiotensin-aldosterone system. These neurohormonal systems also affect the cerebral blood flow, however, discussion regarding these systems seems lacking.

REPLY: We agree with your comments. As you suggested we have added a new paragraph for discussion of several neurohormonal systems, including sympathetic nerve systems and the renin-angiotensin-aldosterone system (P6 L225-239). 

We have added

3.5 Neurohormonal systems of HF

It is known that sympathetic nerve activity and renin-angiotensin-aldosterone system are enhanced in HF patients [53]. The compensatory homeostatic responses to a fall in cardiac output are activation of the sympathetic nervous system and the renin-angiotensin-aldosterone system. This neurohormonal activation is likely beneficial for preserving the blood flow of the brain and other organs in the acute phase of HF. However, with chronic activation, these responses may result in deteriorated effects on the cardiovascular function and morphology including cerebrovascular tissues, possibly leading to reduced CBF in chronic HF patients. Thus, paradoxically, neurohormonal activation to preserve CBF in chronic HF patients is recognized as the most important pathophysiology underlying the progression of HF. Consequently, current pharmacological therapies are targeted to block these neurohormonal activities, in addition to diuretics. However, the direct effects of neurohormonal systems on cerebral circulation and their treatment effects by drugs are still unknown in chronic HF patients as described above.”

#3 In each paragraph, the authors presented several papers regarding some specific cardiac function and cerebral blood flow. After the review of these papers, the authors stated their opinion regarding each specific relationship. For example, they used “may be due to (line 120)” or “clearly (line 167). However, these notions seem not always follow the results of the papers cited, rather, reflect the opinions of the authors. It is recommended the authors should discriminate their opinion from the results of the references.

REPLY: We apologize for these confusing writings. As your suggested, we have re-written to review previous papers and distinguished the results and our opinion clearly for readers (P4 L118-122, P4 L137-143, P5 L181-182, P7 L266).

Minor concerns

#1 Regarding lines 92-99,. The authors stated arterial baroreflex function initiate tachycardia as a response to reduced cardiac function. In general, higher heart rate is a worse prognostic indicator for heart failure. In addition, beta-blockers or ivabradine, both make heart rate slow, are widely used in patients with heart failure with reduced ejection fraction. Please discuss the effect of these drugs on the cerebral blood perfusion.

REPLY: Thank you so much for this constructive comment. As you suggested, we have added the discussion regarding these medications (P4 L143-P5 L164).

We have added

“Treatment with beta blockers reduces the risk of death and the combined risk of death or hospitalization in patients with HF with reduced ejection fraction (HFrEF), and thus it is widely used in HFrEF patients [20]. While it is known that beta blockers improve LVEF, their effects on baroreflex function through the autonomic nervous system may affect brain autoregulation.  However, previous physiological studies [33, 34] showed that beta blockers did not impair arterial-cardiac baroreflex at rest and during exercise, while vagal blockade did impair it, Thus, beta blockers may have beneficial effects on cerebral autoregulation, especially in patients with LVEF improved by beta blockers. More recently, ivabradine was suggested to be used for HF patients with sinus rhythm and a heart rate of ≥70 bpm at rest, in order to reduce hospitalizations and cardiovascular death [20].  Recent animal studies indicated that ivabradine affects neither cardiovascular autonomic control nor arteria baroreflex function [35, 36]. Thus, heart rate control therapies by these drugs are not likely to impact cerebral autoregulation via baroreflex function.”

#2 Regarding lines 105-121 The authors discussed the effect of cardiopulmonary baroreflex on cerebral blood flow. The component of cardiopulmonary baroreflex, blood volume and peripheral vascular resistance, are affected significantly by treatment for heart failure, i.e. diuretics or renin-angiotensin-aldosterone system inhibitors. Please discuss the role of these drugs in regard to cardiopulmonary baroreflex.

REPLY: Thank you so much for this constructive comment. As you suggested, we have added the discussion regarding these drugs (P5 L182-196).

We have added

“In patients with HF who have fluid retention, diuretics such as loop and thiazide are recommended to relieve congestion, improve symptoms, and prevent worsening HF [20]. Since these drugs aim at blood volume reduction to improve congestion, they may increase the risk of severe cerebral hypo-perfusion given that cardiopulmonary baroreflex is impaired in HF patients. However, since the treatment effects of these drugs on cardiopulmonary baroreflex itself are still unknown, further studies would be warranted. Renin-angiotensin system inhibition with ACEi, ARB, or ARNi may also affect cardiopulmonary baroreflex by reducing blood volume and vasoconstriction. Physiologically, this treatment directly exacerbates cardiopulmonary baroreflex through impaired vasoconstrictive ability via blockade of the renin-angiotensin system, leading to worse CBF regulation. However, one previous study suggested that the treatment of HF patients by ACEi improved the outflow of sympathetic nerve activity in the cardiopulmonary baroreflex pathway [48]. The gross effects of renin-angiotensin system inhibition on CBF regulation are still unknown and future studies would be warranted.”

Round 2

Reviewer 1 Report

Thank you for your comments. I have no further questions

Reviewer 2 Report

The authors sufficiently replied to my concerns. Future studies to elucidate complex effects of mechanical and neurohormonal circulatory systems on cerebral blood flow are needed.